# LEARNABLE SPIKING NEURAL P SYSTEM WITH INTERVAL EXCITATION

## ABSTRACT

Spiking Neural P (SN P) systems are parallel distributed models developed by mimicking bio-nervous systems. Past decades have emerged a lot of efforts on theoretical characterizations and modeling plasticity of SN P systems; however, it still remains challenging that interacts with real-world environments due to the limited expressive capacity and the non-differentiable nature of their excitation mechanism. This paper proposes a Learnable Spiking Neural P System with Interval Excitation (LSNP_IE) for real-valued processing. The proposed LSNP_IE employs an interval excitation mechanism and a potential adjustment module, which improve modeling plasticity and enable excitation stability, respectively. The whole system can be adjusted by surrogate gradients beyond hardware. Experimental results conducted on real-world datasets show that LSNP_IE achieves competitive performance compared to traditional non-spiking and spiking models. Our investigations not only reveal the potential of integrating spiking computations with parallel distributed frameworks, but also support the development of hardware-adapted learning.

## 1 INTRODUCTION

Spiking Neural P (SN P) systems are a type of natural computing model that integrates spiking processing with membrane computing (i.e., P systems) (Ionescu et al., 2006; Song et al., 2016). SN P systems simulate the event-driven communication of biological neurons using spikes, while leveraging the parallel and distributed architecture of P systems to enhance computational expressiveness (Binder et al., 2007; Cabarle et al., 2015a;b; Chen et al., 2008; Paun, 2007; Wang et al., 2010). Consequently, SN P systems can be implemented well on neuromorphic hardware, providing further improvements in energy efficiency over conventional CPUs and GPUs (Păun et al., 2006). In recent years, SN P systems have gained increased attention in brain-inspired computing and neuromorphic system design, gradually extending into practical applications such as image recognition (Păun, 2000), graph processing (Roy et al., 2019), and brain-computer interface research (Zhang, 2024).

Currently, SN P systems rely heavily on manually constructed rules, including neuron connections and excitation conditions (Wang et al., 2016), without support for automatic optimization methods, such as backpropagation in deep learning (Wang & Peng, 2013) or spike-timing-dependent plasticity in spiking neural networks (Tavanaei & Maida, 2019). The practical success of SN P systems and its variants usually demands extensive expert knowledge, thus struggling to scale to complex problems and large datasets. In recent years, several efforts have emerged to tackle this challenge, such as enhancing modeling plasticity using weighted SN P systems on simple classification tasks (Ermini & Zandron, 2024; Pleșa et al., 2024; Zhang et al., 2022) and developing heuristic optimization techniques including Hebbian learning (Song et al., 2019) and evolutionary algorithms (Dong et al., 2021). However, SN P systems still lack a unified and flexible learning framework, making it difficult to achieve adaptive learning in real-world environments.

In this paper, we propose the Learnable Spiking Neural P System with Interval Excitation (LSNP_IE) for neuromorphic image classification. The proposed LSNP_IE consists of a feedforward perceptron-based network topology, an interval excitation mechanism, and a potential adjustment module, as illustrated in Figure 1. These components together enhance the expressive capacity of LSNP_IE. In addition, LSNP_IE supports joint training with surrogate gradients. Experimental results conducted on neuromorphic image datasets demonstrate our method's effectiveness.

Our main contributions are summarized as follows.

- We propose LSNP_IE, which integrates an interval excitation mechanism and a potential adjustment module into SN P systems, thereby enhancing their expressive capacity.
- We introduce a surrogate gradient-based backpropagation algorithm, enabling end-to-end training and adaptive learning for LSNP_IE on large-scale neuromorphic datasets.
- Experimental results conducted on neuromorphic image datasets show that our method achieves competitive performance compared to traditional non-spiking and spiking models.

The remainder of this paper is organized as follows. Section 2 reviews related work on SN P systems. Section 3 introduces the proposed LSNP_IE model. Section 4 presents the experiments, and Section 5 concludes the paper.

## 2 RELATED WORK

SN P systems are a natural computing model that integrates Spiking Neural Networks (SNNs) and membrane computing (P systems). They were initially proposed by Ionescu et al. (2006), aiming to simulate spike-based, event-driven communication between neurons while enhancing computational expressiveness through the parallel and distributed structures of membrane computing. While both models employ spikes, SN P systems fundamentally differ from SNNs in their theoretical underpinnings and structural paradigms. SNNs focus on modeling biological neuron dynamics, whereas SN P systems, originating from the more computation-oriented membrane computing theory, possess an inherently parallel, distributed, and modular structure. Unlike monolithic SNNs, the clear separation of *membranes* and *rules* in SN P systems facilitates modular hardware design, paving a unique path toward efficient, low-energy implementations. Early research on SN P systems primarily focused on foundational aspects such as model definitions, syntax, excitation rules, and Turing completeness (Păun et al., 2006). Subsequently, various system variants, including those with and without delays, were developed, and their capabilities in formal language recognition, recursive function simulation, and complexity analysis were explored in depth (Leporati et al., 2022). SN P systems have garnered increasing attention in fields such as brain-inspired computing (Zahra et al., 2022) and hardware-software co-design (Zhang et al., 2024), gradually expanding into practical applications like image recognition (Song et al., 2019) and graph mining (Bai et al., 2025). The study of SN P systems holds profound theoretical and practical significance. Theoretically, as a Turing-complete model, they provide a novel modeling approach with temporal and parallel characteristics, pushing the boundaries of formal computational models. In brain-inspired intelligence, the spiking mechanism of SN P systems closely aligns with biological neuronal dynamics, offering new modeling tools for neuromorphic computing and brain-computer interface research (Zhang, 2024). Additionally, their inherent parallelism and distributed structures align well with modern multi-core computing architectures (Odasco et al., 2023). Currently, research on SN P systems is progressing toward integration with deep learning, the development of efficient simulators, and hardware-software co-design, showing strong interdisciplinary potential and broad applicability.

For classification problems, Wang et al. (2010) proposed the weighted SN P system, introducing adjustable synaptic weights to enable dynamic regulation of spike transmission intensity. This approach laid the groundwork for integrating SN P systems with machine learning optimization methods. More details about the concept of weighted SN P systems are provided in AppendixA.1. Subsequently, Wang et al. (2016) realized an SN P system with 87 neurons capable of computing any Turing-computable recursive function, envisaging the application of SN P systems in learning mechanisms and image classification. Following this, Song et al. (2019) employed simple Hebbian learning functions in an SN P system to recognize alphabetic digits, demonstrating its potential in pattern recognition tasks. As inherently parallel systems, SN P systems also offer significant energy efficiency advantages (Odasco et al., 2023), highlighting their applicability to learning-based classification tasks. Building on these foundational works, several variants of SN P systems have been proposed to improve classification performance, including approaches incorporating hierarchical structures (Zhang et al., 2022), fuzzy logic (Wang & Peng, 2013), modularity (Ermini & Zandron, 2024), and modified learning rules (Cabarle et al., 2015a). These methods have enabled applications ranging from nonlinear classification of multidimensional data to malware, phishing, spam detection, and multiclass recognition tasks. However, they often exhibit limitations such as

scalability issues with high-dimensional data, difficulty tuning synaptic strengths via gradient-based learning, and restricted temporal modeling capabilities. Dong et al. (2024) proposed a learnable numerical SN P system, introducing arbitrary numerical variables and production functions to overcome the discontinuities of traditional SN P systems. The system was validated on eight benchmark datasets from machine learning repositories, confirming its feasibility and effectiveness. However, its dimensionality-lifting structure suffers from rapidly increasing computational complexity in high-dimensional scenarios and still fails to address inter-layer learning effectively. The LSNP_IE proposed in this paper effectively addresses these limitations, as it can process large-scale temporal data and support gradient-based joint training across multiple layers.

## 3 LEARNABLE SPIKING NEURAL P SYSTEM WITH INTERVAL EXCITATION

This section introduces the LSNP_IE, as illustrated in Figure 1, which consists of the network topology, interval excitation mechanism, potential adjustment module, and back-propagation.

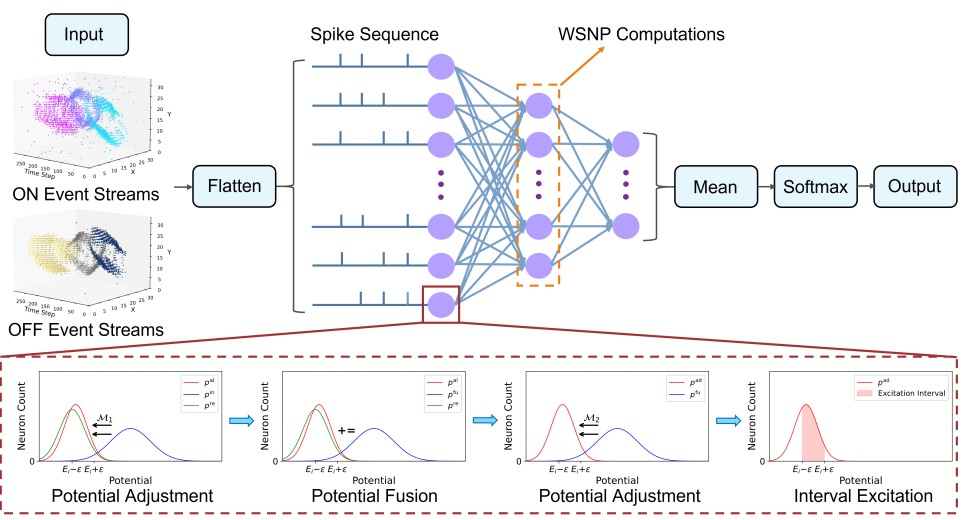

Figure 1: The workflow of LSNP_IE.

### 3.1 NETWORK TOPOLOGY

The proposed LSNP_IE employs the Multi-Layer Perceptron (MLP) architecture. The propagation computation between two adjacent layers follows the feed-forward Weighted Spiking Neural P (WSN P) system (Wang et al., 2010), which is an extended model of SN P systems equipped with synaptic weights. Consider the $l$-th layer with $m_l$ neurons for $l \in [L]$ and $m_l \geq 1$; the feed-forward WSN P system of degree $m_l$ is a construct of the form $\Pi^{(l)} = (\sigma_1, \sigma_2, \ldots, \sigma_{m_l}, syn, in, out)$, where

1. $\sigma_i = (p_i, R_i)$ $(1 \leq i \leq m_l)$ denotes the $i$-th neuron, where
   - $p_i(t) \in \mathbb{R}^+$: the potential value of the $i$-th neuron at timestamp $t$;
   - $R_i(p_i)$: a finite set of excitation rules, in the form $R_i : p_i(t) \mapsto 1$.

2. $syn \subseteq [m_{l-1}] \times [m_l] \times \mathbb{R}^+$ is the set of weighted synapses, where $(i, j, w_{ij})$ denotes a synapse from neuron $\sigma_i$ in layer $l-1$ to neuron $\sigma_j$ in layer $l$ with synaptic weight $w_{ij}$, It is required that $i \neq j$, $w_{ij} \neq 0$, and each pair $(i, j)$ appears at most once in $syn$.

3. $in \subseteq [m_{l-1}]$ and $out \subseteq [m_l]$ denote the disjoint sets of input channels and output neurons, where input channels receive spike trains from the environment or equivalently the previous layer and output neurons emit spikes to the environment or equivalently the next layer.

The propagation computations of WSN P systems enable dynamic regulation between neurons and provide a mathematical foundation for system parameter optimization and parallel distributed computing (Odasco et al., 2023).

## 3.2 INTERVAL EXCITATION MECHANISM

Existing SN P systems usually employ the point-triggered mechanism where the rules can only be triggered when the potential $p_i$ is exactly equal to the excitation threshold $E_i$ for neuron $\sigma_i$. In other words, $R_i$ comprises one excitation rule of $p_i(t) \to 1$ if and only if $p_i = E_i$, which contributes to integer-valued computations (Wang et al., 2010). However, when one applies the SN P system and its variants to environments with real-valued or floating-point numbers, such point-triggered mechanisms are overly strict when triggering neuron rules, directly causing interruptions in information flow, since the continuous probability of neuron excitation approaches zero, i.e., $P(p_i = E_i) \approx 0$.

In this work, we propose an interval excitation mechanism with $\epsilon$-neighborhood, which extends the triggering domain from a single point $E_i$ to an interval $[E_i - \epsilon, E_i + \epsilon]$ where $\epsilon$ denotes the threshold tolerance. The proposed interval excitation mechanism can be formulated as follows.

- If $p_i(t) < E_i - \epsilon$, then $p_i(t)$ is reset to zero.
- If $p_i(t) > E_i + \epsilon$, then $p_i(t)$ remains unchanged.
- If $E_i - \epsilon \leq p_i(t) \leq E_i + \epsilon$, then
  1. A rule $E_i/d \to 1$ is randomly selected and applied from the rule set $R_i$, where $d$ denotes potential decay coefficient.
  2. The neuron potential decreases to $E_i - d$.
  3. A unit potential is emitted through the weighted synapse to the connected neurons.

To provide a concise formulation of the interval excitation process, we introduce the potential emission function $\mathcal{F}(x)$ and the potential update function $\mathcal{G}(x)$, which represent the dynamics of the emitted potential and the updated potential after excitation, respectively. As shown in Figure 2(a) and Figure 2(c), $\mathcal{F}(x)$ and $\mathcal{G}(x)$ are separately defined as follows

$$\mathcal{F}(x) = \begin{cases} 1, & |x - E_i| \leq \epsilon, \\ 0, & \text{otherwise}. \end{cases} \quad \text{and} \quad \mathcal{G}(x) = \begin{cases} 0, & x < E_i - \epsilon, \\ x - d, & E_i - \epsilon \leq x \leq E_i + \epsilon, \\ x, & x > E_i + \epsilon. \end{cases} \quad (1)$$

This modification enables LSNP_IE to better handle data with real-valued and floating-point formats, improving operational smoothness and robustness.

## 3.3 POTENTIAL ADJUSTMENT MODULE

Here, we introduce a potential adjustment module, defined as

$$\mathcal{M}(\boldsymbol{x}) = \gamma \, \frac{\boldsymbol{x} - \mu}{\varsigma + \delta} + \beta, \quad (2)$$

where $\boldsymbol{x}$ denotes the received signals, $\mu$ and $\varsigma$ are the mean and standard deviation of $\boldsymbol{x}$, $\gamma$ and $\beta$ are preset scale and shift parameters, whose specific settings are listed in Table 1, and $\delta = 10^{-8}$ is a small constant ensuring numerical stability.

Figure 1 illustrates the integration of the potential adjustment module and the workflow of LSNP_IE. As shown, each neuron contains two such modules. The input potential $p^{\text{in}}$ is first processed by the first potential adjustment module ($\mathcal{M}_1$), yielding the aligned potential $p^{\text{al}}$, which are aligned with the residual potential $p^{\text{re}}$. Subsequently, the aligned potential $p^{\text{al}}$ and the residual potential $p^{\text{re}}$ undergo potential fusion, resulting in the fused potential $p^{\text{fu}}$. The fused potential is then passed through the second potential adjustment module ($\mathcal{M}_2$) to produce the adjusted potentials $p^{\text{ad}}$, effectively shifting the distribution toward the excitation interval. Finally, the adjusted potential $p^{\text{ad}}$ is subjected to interval excitation, as detailed in Section 3.2.

$\mathcal{M}_1$ addresses the distribution mismatch that may occur between $p^{\text{in}}$ and $p^{\text{re}}$, and prevents the useful information in $p^{\text{re}}$ from being overwhelmed during potential fusion. $\mathcal{M}_2$ shifts potentials toward the excitation interval, thereby enabling excitation stability. We also conduct ablation experiments in Section 4.3 and Appendix A.2 to verify the empirical effects of the two potential adjustment modules. Note that for the input layer, only positive potentials are adjusted due to its sparsity.

## 3.4 BACK-PROPAGATION

This subsection presents the training computations with back-propagation and surrogate gradients, which enable the proposed LSNP_IE to achieve learning flexibility. Here, we adopt the supervised learning paradigm. Given the target $y$, we formulate an optimization problem by minimizing the loss function $\mathcal{L}(\hat{y}, y)$, where $\hat{y}$ denotes the output of LSNP_IE. It is recommended to use the cross-entropy loss with one-hot label for classification tasks and the square loss for regression tasks.

For any synaptic weight $w_{ij}$, its update amount is computed from the gradient according to

$$\frac{\partial \mathcal{L}(\hat{y}, y)}{\partial w_{ij}} = \sum_{t=1}^{T} \frac{\partial \mathcal{L}(\hat{y}, y)}{\partial \hat{y}} \frac{\partial \hat{y}}{\partial R_j(p_j^{\mathrm{ad}})} \frac{\partial R_j(p_j^{\mathrm{ad}})}{\partial p_j^{\mathrm{ad}}} \frac{\partial p_j^{\mathrm{ad}}}{\partial p_j^{\mathrm{fu}}} \frac{\partial p_j^{\mathrm{fu}}}{\partial p_j^{\mathrm{al}}} \frac{\partial p_j^{\mathrm{al}}}{\partial p_j^{\mathrm{in}}} \frac{\partial p_j^{\mathrm{in}}}{\partial w_{ij}} \,, \tag{3}$$

where $T$ denotes the maximum number of timestamps, $p_j^{\cdot}$ represents various potentials of the $j$-th neuron, and the time index $t$ is omitted for brevity. The term $\partial \mathcal{L}(\hat{y}, y)/\partial \hat{y}$ represents the gradient of the loss function $\mathcal{L}(\hat{y}, y)$ with respect to the predicted output $\hat{y}$, and $\partial \hat{y}/\partial R_j(p_j^{\mathrm{ad}})$ denotes the gradient of $\hat{y}$ with respect to the excitation output $R_j(p_j^{\mathrm{ad}})$. In addition, $\partial p_j^{\mathrm{fu}}/\partial p_j^{\mathrm{al}}$ and $\partial p_j^{\mathrm{in}}/\partial w_{ij}$ represent the gradients of the fused potentials $p_j^{\mathrm{fu}}$ with respect to the aligned potentials $p_j^{\mathrm{al}}$ and of the input potentials $p_j^{\mathrm{in}}$ with respect to the synaptic weight $w_{ij}$, respectively.

The term $\partial p_j^{\mathrm{ad}}/\partial p_j^{\mathrm{fu}}$ denotes the gradient of the adjusted potentials $p_j^{\mathrm{ad}}$ with respect to the fused potentials $p_j^{\mathrm{fu}}$. Similarly, $\partial p_j^{\mathrm{al}}/\partial p_j^{\mathrm{in}}$ denotes the gradient of the aligned potentials $p_j^{\mathrm{al}}$ with respect to the input potentials $p_j^{\mathrm{in}}$. Both of them can be derived based on Eq. (2), yielding

$$\begin{cases} \dfrac{\partial p_j^{\mathrm{ad}}}{\partial p_j^{\mathrm{fu}}} = \dfrac{\partial \mathcal{M}(p_j^{\mathrm{fu}})}{\partial p_j^{\mathrm{fu}}} = \dfrac{\varsigma\gamma\,(m_l - 1)(\varsigma + \delta) - \gamma\,(p_j^{\mathrm{fu}} - \mu)^2}{\varsigma m_l\,(\varsigma + \delta)^2} \,, \\[4mm] \dfrac{\partial p_j^{\mathrm{al}}}{\partial p_j^{\mathrm{in}}} = \dfrac{\partial \mathcal{M}(p_j^{\mathrm{in}})}{\partial p_j^{\mathrm{in}}} = \dfrac{\varsigma\gamma\,(m_l - 1)(\varsigma + \delta) - \gamma\,(p_j^{\mathrm{in}} - \mu)^2}{\varsigma m_l\,(\varsigma + \delta)^2} \end{cases}$$

The term $\partial R_j(p_j^{\mathrm{ad}})/\partial p_j^{\mathrm{ad}}$ denotes the gradient of the excitation output $R_j(p_j^{\mathrm{ad}})$ with respect to the adjusted potentials $p_j^{\mathrm{ad}}$. Based on the interval excitation mechanism, it is given as

$$\frac{\partial R_j(p_j^{\mathrm{ad}})}{\partial p_j^{\mathrm{ad}}} = \frac{\partial R_j(p_j^{\mathrm{ad}})}{\partial \mathcal{F}(p_j^{\mathrm{ad}})} \frac{\partial \mathcal{F}(p_j^{\mathrm{ad}})}{\partial p_j^{\mathrm{ad}}} + \frac{\partial R_j(p_j^{\mathrm{ad}})}{\partial \mathcal{G}(p_j^{\mathrm{ad}})} \frac{\partial \mathcal{G}(p_j^{\mathrm{ad}})}{\partial p_j^{\mathrm{ad}}} \,, \tag{4}$$

Note that $R_j$ is non-differentiable, as evident from the functions $\mathcal{F}$ and $\mathcal{G}$ defined in Eq. (1), respectively. To solve Eq. (4), we employ the surrogate gradient (SG) approach (Che et al., 2022; Li et al., 2021; Neftci et al., 2019; Wu et al., 2018),

$$\frac{\partial R_j(\mathcal{M}(p_j^{\mathrm{ad}}))}{\partial \mathcal{M}(p_j^{\mathrm{ad}})} \approx SG \,.$$

**Surrogate Gradients.** To address the non-differentiability of $R_j$, we adopt the surrogate gradient approach to approximate the derivatives of $\mathcal{F}$ and $\mathcal{G}$ with suitable continuous functions.

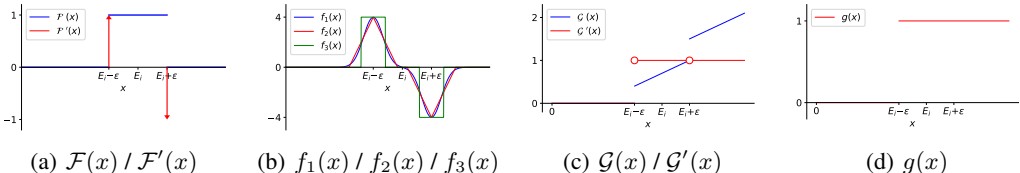

(a) $\mathcal{F}(x)$ / $\mathcal{F}'(x)$    (b) $f_1(x)$ / $f_2(x)$ / $f_3(x)$    (c) $\mathcal{G}(x)$ / $\mathcal{G}'(x)$    (d) $g(x)$

Figure 2: Surrogate gradient functions used for the potential emission and potential update functions.

First, the theoretical derivative of $\mathcal{F}$ is given by the Dirac function $\delta(x)$ in Figure 2(a),

$$\mathcal{F}'(x) = \delta(x - (E_i - \epsilon)) - \delta(x - (E_i + \epsilon)) \,.$$

Here, we employ three surrogate gradients to approximate $\mathcal{F}'(x)$, including a bimodal Gaussian surrogate $f_1$, a piecewise linear surrogate $f_2$, and a rectangular pulse surrogate $f_3$, as illustrated in Figure 2(b) and defined as

$$
\begin{cases}
f_1(x) = \dfrac{1}{\sqrt{2\pi}\sigma_1}\left[\exp\left(-\dfrac{(x-(E_i-\epsilon))^2}{2\sigma_1}\right) - \exp\left(-\dfrac{(x-(E_i+\epsilon))^2}{2\sigma_1}\right)\right]\,, \\[2mm]
f_2(x) = \begin{cases} -16|x-(E_i-\epsilon)| + 4\,, & |x-(E_i-\epsilon)| \le 0.25\,, \\ 16|x-(E_i+\epsilon)| - 4\,, & |x-(E_i+\epsilon)| \le 0.25\,, \\ 0\,, & \text{otherwise}\,, \end{cases} \\[4mm]
f_3(x) = \begin{cases} 4\,, & |x-(E_i-\epsilon)| \le 0.125\,, \\ -4\,, & |x-(E_i+\epsilon)| \le 0.125\,, \\ 0\,, & \text{otherwise}\,, \end{cases}
\end{cases}
$$

where $\sigma_1 = 0.01$. All surrogate gradients satisfy $\int_{-\infty}^{+\infty} |f_i(x)|\,dx = 2$ to ensure consistency with $\delta(x)$. The effects of three surrogate gradients on model performance are investigated in Section 4.3.

For $\mathcal{G}$, the discontinuity at $x = E_i \pm \epsilon$ is negligible, as illustrated in Figure 2(c). Since the probability of a value falling exactly at these points in the continuous domain is virtually zero, we adopt a surrogate formulation $g$, which is illustrated in Figure 2(d) and defined as

$$
g(x) = \begin{cases} 0\,, & x < E_i - \epsilon\,, \\ 1\,, & x \ge E_i - \epsilon\,. \end{cases}
$$

Therefore, Eq. (3) can be implemented. Note that Eq. (4) is intended to detail the gradient propagation path within a single neuron through our novel modules, such as the interval excitation and potential adjustment. As for the inter-layer gradient propagation, it is encapsulated in the term $\partial \hat{y}/\partial R_j(p_j^{ad})$. This term represents the accumulated gradient propagated backward from the final output $\hat{y}$ to the output of neuron $j$, $R_j$, thereby containing the gradient information from all subsequent layers. As the network's gradient calculation follows the standard chain rule, this training framework inherently supports network architectures of arbitrary depth. The detailed training procedure for LSNP_IE is listed in Algorithm1.

---

**Algorithm 1** Training Procedure of LSNP_IE.

---

**Input:** Training set $D = \{(x_k, y_k)\}_{k=1}^{m}$, learning rate $\eta$, neuron counts $m_l$ for $l \in [L]$
**Output:** Final weights $W^{(l)}$ for $l \in [L]$
**Procedure:**
 1: **Initialize weights:** $W^{(l)} \sim \mathcal{N}(0, \sqrt{2/(m_{l-1} + m_l)})$ for $l \in [L]$
 2: **repeat**
 3:    **Forward propagation:**
 4:    **for** $t = 1$ to $T$ **do**
 5:        All neurons perform parallel potential emission and update
 6:    **end for**
 7:    **Backward propagation:**
 8:    **for** $t = T$ to $1$ **do**
 9:        Compute $\partial \mathcal{L}/\partial W^{(l)}$ using Eq. (3) for $l \in [L]$
10:    **end for**
11:    Update $W^{(l)}$ using $\partial \mathcal{L}/\partial W^{(l)}$ for $l \in [L]$
12: **until** stopping condition is met

---

## 4 EXPERIMENTS

This section conducts experiments to demonstrate the effectiveness of the proposed LSNP_IE.

### 4.1 CONFIGURATIONS

We evaluate our models on two neuromorphic datasets, namely N-MNIST and MNIST-DVS (Orchard et al., 2015), whose sample images are shown in Appendix A.3. The N-MNIST dataset con-

tains 60,000 training and 10,000 test samples, each represented as an event stream of $(x, y, t, p)$ with $x, y \in [0, 34]$ and $p \in \{0, 1\}$. For our experiments, the event streams are discretized into 300 time frames with a duration of $1,000\,\mu$s each. From the training set, 54,000 samples are used for training and 6,000 for validation. Model training is performed over 50 epochs. The batch size and learning rate are determined via grid search, yielding optimal values of 256 and 0.01, respectively. The MNIST-DVS dataset comprises 30,000 samples, evenly divided into three spatial scales (commonly named scale 4, 8, and 16), with 10,000 samples per scale. Each sample is encoded as an event stream with $x, y \in [0, 127]$ and $p \in \{0, 1\}$, and is discretized into 500 time frames, each with a duration of $5,000\,\mu$s. For each scale, the dataset is split into 8,000 for training, 1,000 for validation, and 1,000 for testing. Independent models are trained for each scale using a batch size of 64, a learning rate of 0.01, and 20 training epochs. For consistency across datasets, we adopt identical core hyperparameters: the excitation threshold is set to $E_i = 1.2$, threshold tolerance $\epsilon = 0.3$, and potential decay coefficient $d = 0.6$. All weights are initialized using Xavier initialization (Glorot & Bengio, 2010) and updated via the Adam optimizer (Kingma & Ba, 2014) with gradient clipping (Pascanu et al., 2013). The specific hyperparameters are listed in Table 1. Each configuration is evaluated across five runs to ensure result robustness. The output neuron with the highest membrane potential is selected as the predicted output. Here, we employ several non-spiking and spiking models as baseline methods. All models are trained on an NVIDIA RTX 4090 GPU with 24GB VRAM and 120GB system memory, requiring approximately 4 hours per model.

Table 1: Hyperparameter settings for N-MNIST and MNIST-DVS.

| Parameters | N-MNIST | MNIST-DVS |
|---|---|---|
| Batch size | 256 | 64 |
| Learning rate $\eta$ | 0.01 | 0.01 |
| Excitation threshold $E_i$ | 1.2 | 1.2 |
| Threshold tolerance $\epsilon$ | 0.3 | 0.3 |
| Potential decay coefficient $d$ | 0.6 | 0.6 |
| Shift parameter $\beta$ $(l \in [L-1])$ | 1 | 1 |
| Scale parameter $\gamma$ $(l \in [L-1])$ | $\sqrt{0.1}$ | $\sqrt{0.1}$ |
| Shift parameter $\beta$ $(l = L)$ | 0 | 0 |
| Scale parameter $\gamma$ $(l = L)$ | $\sqrt{0.5}$ | $\sqrt{0.5}$ |
| Network architecture | 2312-500-500-10 | 32768-500-500-10 |
| Number of time frames | 300 | 500 |

### 4.2 EXPERIMENTAL RESULTS

**N-MNIST.** Table 2 lists the accuracy of LSNP_IE and baseline methods on the N-MNIST dataset, where the best performance is marked in bold. It is observed that SLAYER, the typical spiking MLP method, achieves the best performance. In comparison, our proposed LSNP_IE model outperforms both traditional MLP and SKIM, with only a 0.98% gap from SLAYER. These results demonstrate that LSNP_IE achieves competitive performance against traditional spiking and non-spiking models.

We plot the training and validation accuracy curves throughout the training process on the N-MNIST dataset in Figure 3(a). It can be observed that our model achieves over 95% accuracy on both the training and validation sets after only two epochs. Furthermore, the model consistently exhibits stable convergence during training, with the validation accuracy steadily increasing and eventually reaching a peak of 98.10%. The training and validation curves remain closely aligned throughout, implying not only stable learning, but also strong generalization performance. In conclusion, the experimental results show that LSNP_IE has fast and stable convergence characteristics.

**MNIST-DVS.** Table 2 lists the accuracy of LSNP_IE and baseline methods on the MNIST-DVS dataset, where the best performance is marked in bold. It is clear that LSNP_IE delivers robust performance across different input scales. Notably, LSNP_IE achieves the best accuracy at 99.80%, outperforming all listed baselines. Although accuracy comparisons may be influenced by variations in data splits and ambiguous reporting of input scales across studies, the overall results confirm the adaptability and promise of LSNP_IE for neuromorphic computing tasks.

We also plot the training and validation accuracy curves during the training process on the MNIST-DVS dataset at scale 8 in Figure 3(b). It is observed that LSNP_IE achieves over 98% accuracy on both the training and validation sets by the fourth epoch. Moreover, LSNP_IE shows stable convergence, with validation accuracy continually rising and eventually reaching a peak of 98.60%. The training and validation curves remain closely aligned throughout, reflecting rapid and stable convergence as well as strong generalization capability. In summary, LSNP_IE not only exceeds prior non-spiking, spiking MLP and spiking CNN models in accuracy, but also shows desirable convergence properties.

Table 2: Accuracy of the proposed LSNP_IE and baseline methods on N-MNIST and MNIST-DVS.

| Datasets | Methods | Type | Accuracy (%) |
|---|---|---|---|
| N-MNIST | MLP (Lee et al., 2016) | Non-spiking MLP | 97.80 |
| | SKIM (Cohen et al., 2016) | Spiking MLP | 92.87 |
| | STBP (Wu et al., 2018) | Spiking MLP | 98.78 |
| | HM2-BP (Jin et al., 2018) | Spiking MLP | 98.84 |
| | SLAYER (Shrestha & Orchard, 2018) | Spiking MLP | **98.89** |
| | LSNP_IE (this work) | Spiking MLP | $97.91 \pm 0.04$ |
| MNIST-DVS | SSFE (Fang et al., 2024) | Non-spiking CNN | 99.10 |
| | LIAF-Net (Wu et al., 2021) | Spiking CNN | 99.10 |
| | GEM-SNN (Jang & Simeone, 2022) | Spiking MLP | 97.20 |
| | SCTFA (Cai et al., 2023) | Spiking CNN | 98.70 |
| | LSNP_IE (scale4, this work) | Spiking MLP | $99.02 \pm 0.28$ |
| | LSNP_IE (scale8, this work) | Spiking MLP | $\mathbf{99.60 \pm 0.20}$ |
| | LSNP_IE (scale16, this work) | Spiking MLP | $97.48 \pm 0.18$ |

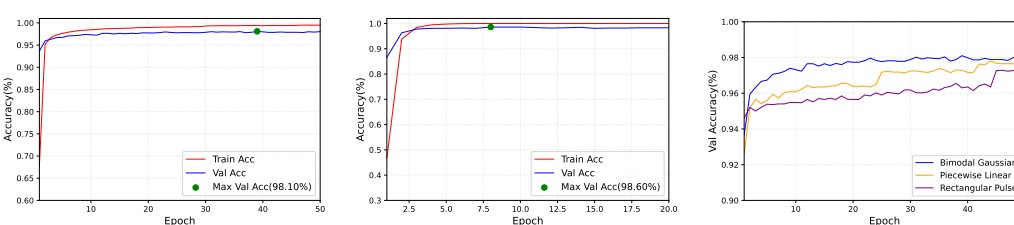

(a) Accuracy curve (N-MNIST) (b) Accuracy curve (MNIST-DVS) (c) Accuracy of Surrogate Gradients

Figure 3: Accuracy curves for the (a) N-MNIST and (b) MNIST-DVS datasets, while (c) accuracy curves of the three Surrogate Gradients.

### 4.3 VERIFICATION OF ALTERNATIVE CONFIGURATIONS

This subsection conducts experiments on the N-MNIST dataset to investigate the effects of different configurations on the performance of LSNP_IE.

**Weight Initialization.** Here, we employ two typical weight initialization techniques, namely Xavier initialization (Glorot & Bengio, 2010) and uniform initialization ($U[-1, 1]$), as commonly used in prior studies (Jin et al., 2018; Wu et al., 2018). Table 3 shows that Xavier initialization achieves higher accuracy than uniform initialization. Therefore, we recommend Xavier initialization to initialize the weights.

**Potential Decay Coefficient $d$.** Here, we investigate the effect of the potential decay coefficient $d$ by conducting an ablation study across several typical values of 0.3, 0.4, 0.5, 0.6, 0.7, and 0.8. Table 4 summarizes the results, showing that the highest accuracy is obtained with $d = 0.6$. Performance noticeably degrades when $d$ deviates from this optimal value in either direction. Specifically, larger values such as $d = 0.8$ tend to induce excessive decay, causing insufficient potential accumulation and information loss, while smaller values, such as $d = 0.3$, result in overly persistent membrane

Table 3: Accuracy of two weight initializations.

| Methods | Accuracy |
|---------|----------|
| Uniform Initialization | 97.03% |
| Xavier Initialization | **97.93%** |

Table 5: Ablation study on $\mathcal{M}_1$.

| Methods | Accuracy |
|---------|----------|
| Without $\mathcal{M}_1$ | 97.49% |
| With $\mathcal{M}_1$ | **97.93%** |

Table 4: Accuracy of different $d$ values.

| $d$ | Accuracy | $d$ | Accuracy |
|-----|----------|-----|----------|
| 0.3 | 97.03% | 0.6 | **97.93%** |
| 0.4 | 97.04% | 0.7 | 97.49% |
| 0.5 | 97.36% | 0.8 | 97.16% |

Table 6: Accuracy of surrogate gradients.

| Methods | Accuracy |
|---------|----------|
| Bimodal Gaussian | **97.93%** |
| Piecewise Linear | 97.46% |
| Rectangular Pulse | 97.22% |

potential and frequent excitation. Therefore, we recommend setting $d = 0.6$, as it offers the best balance between efficient gradient propagation and biologically plausible membrane potential decay.

**The First Potential Adjustment Module.** Here, we conduct an ablation study comparing LSNP_IE with and without $\mathcal{M}_1$ to assess the effectiveness of the module introduced in Section 3.3. Table 5 shows that applying $\mathcal{M}_1$ improves test accuracy by approximately 0.5%. This result indicates that $\mathcal{M}_1$ effectively harmonizes potential distributions and enhances information integration across timestamps. We conclude that $\mathcal{M}_1$ plays a crucial role in improving overall model performance.

**Surrogate Gradients.** Here, we assess the impact of different surrogate gradients on the training of our model. Specifically, we compare three commonly used candidates—bimodal Gaussian, piecewise linear, and rectangular pulse surrogates. These surrogates approximate the non-differentiable potential emission function $\mathcal{F}$, as detailed in Section 3.4. According to Table 6, the bimodal Gaussian surrogate yields the highest accuracy, outperforming the piecewise linear and rectangular pulse surrogates. We attribute this superior performance to its smooth gradient shape. Unlike the rectangular pulse with its abrupt boundary changes and the piecewise-linear function with its non-smooth kinks, the bimodal Gaussian is continuous and smooth. This property facilitates a more stable gradient descent, helping to prevent oscillations or overshooting in weight updates and thus promoting convergence to a high-quality local optimum. This superiority is further illustrated in Figure 3(c), where the bimodal Gaussian surrogate consistently leads in validation accuracy throughout training. Therefore, we recommend using the bimodal Gaussian surrogate for LSNP_IE.

## 5 CONCLUSIONS AND DISCUSSIONS

In this paper, we proposed LSNP_IE to address key limitations of traditional SN P systems in interacting with real-world environments. Our approach introduces a differentiable training framework through an interval excitation mechanism, a potential adjustment module, and surrogate gradients, enabling robust excitation dynamics. LSNP_IE demonstrates strong performance on large-scale neuromorphic datasets, achieving competitive accuracy on N-MNIST, outperforming non-spiking MLPs and early spiking MLPs while approaching the performance of state-of-the-art methods. LSNP_IE also sets a new benchmark on MNIST-DVS and surpasses all existing approaches. These results highlight LSNP_IE's effectiveness and underscore the potential of SN P systems for scalable, deep learning tasks. To our knowledge, this is the first work to demonstrate the feasibility of applying SN P systems to complex visual classification within a parallel distributed framework, paving the way for hardware-adapted intelligent computation.

While LSNP_IE offers substantial improvements in training flexibility and classification accuracy for SN P systems, there remain key avenues for exploration. For example, integrating convolutional structures or advanced modules from cutting-edge spiking neural networks could further enhance spatial feature extraction and model generalization. In addition, future work will focus on extending LSNP_IE to more challenging and diverse real-world tasks, such as gesture recognition and event-stream video classification, to assess scalability and robustness in dynamic real-world environments.

## ETHICS STATEMENT

The research presented in this paper is purely computational and focuses on the development of an algorithmic model. All experiments were conducted on publicly available and anonymized datasets (N-MNIST and MNIST-DVS), which do not contain any personally identifiable or sensitive information. Our work does not involve human subjects, animal testing, or confidential data. To the best of our knowledge, we foresee no direct negative ethical implications or societal consequences resulting from this research.

## REPRODUCIBILITY STATEMENT

To ensure the reproducibility of our work, this paper fully discloses all necessary details to replicate the main experimental results. Section 4.1 provides a comprehensive description of the training and testing procedures, including dataset specifications and splits, all hyperparameter values and their selection process, optimizer types, and other relevant configuration settings. While the source code is not publicly available at the time of submission, we believe the details provided are sufficient for independent reproduction of our findings. We commit to releasing the complete source code publicly upon acceptance of the paper.

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

# A APPENDIX

## A.1 WEIGHTED SPIKING NEURAL P SYSTEMS

Weighted Spiking Neural P (WSN P) systems (Wang et al., 2010) are an extended form of SN P systems, which introduce synaptic weights to enable dynamic regulation of spike transmission and establish a mathematical foundation for parameter optimization.

### A.1.1 FORMAL DEFINITION

A WSNP system of degree $m \geq 1$ is a construct of the form

$$\Pi = (\sigma_1, \sigma_2, \ldots, \sigma_m, syn, in, out),$$

in which

1. $\sigma_i = (p_i, R_i)$ $(1 \leq i \leq m)$ denotes the $i$-th neuron, where
   - $p_i(t) \in \mathbb{R}^+$: the potential value of the $i$-th neuron at timestamp $t$;
   - $R_i$: a finite set of excitation rules, in the form $E_i/d \rightarrow 1$, where
     - $E_i \in \mathbb{R}^+$, $E_i \geq 1$, which denotes the excitation threshold shared by all rules of $\sigma_i$.
     - $d \in \mathbb{R}^+$, $0 < d \leq E_i$, which denotes the potential decay coefficient.
2. $syn \subseteq [m] \times [m] \times \mathbb{R}^+$ is the set of weighted synapses, where $(i, j, w_{ij})$ denotes a synapse from neuron $\sigma_i$ to neuron $\sigma_j$ with synaptic weight $w_{ij}$, subject to
   - No self-loops: $\forall (i, j, w_{ij}) \in syn \Rightarrow i \neq j$
   - Non-zero weights: $\forall (i, j, w_{ij}) \in syn \Rightarrow w_{ij} \neq 0$
   - Unique connectivity: $\forall (i, j) \in [m]^2$, there exists at most one synapse $(i, j, w_{ij}) \in syn$
3. $in, out \subseteq [m]$ denote the disjoint sets of input and output neurons, where input channels receive spike trains from the environment, while output neurons emit spikes back to it.

### A.1.2 DYNAMIC EVOLUTION MECHANISM

Similar to standard SN P systems, the WSN P system operates under a global clock that tracks discrete timestamps $t = 0, 1, 2, \ldots$. The system evolves step by step until a halting condition is met, which occurs when no neuron has any applicable excitation rules at a given timestamp. At each timestamp $t$, the following phases are executed in sequence.

**Potential Reception** At each timestamp $t$, each neuron receives potentials from the environment or from other neurons. For every synapse $(i, j, w_{ij}) \in syn$, the potential received by neuron $\sigma_j$ at timestamp $t$ is given by

$$\Delta p_j(t) = \sum_{(i,j,w_{ij}) \in syn} w_{ij} \cdot \delta_i(t-1),$$

where $\delta_i(t-1) = 1$ if and only if $\sigma_i$ excites at timestamp $t-1$.

**Rule-based Excitation** Each neuron attempts to excite before the end of timestamp $t$, as follows.

- If $p_i(t) < E_i$, then $p_i(t)$ is reset to zero.
- If $p_i(t) > E_i$, then $p_i(t)$ remains unchanged.
- If $p_i(t) = E_i$, then
  1. A rule $E_i/d \rightarrow 1$ is randomly selected and applied from the rule set $R_i$.
  2. The neuron potential decreases to $E_i - d$.
  3. A unit potential is emitted through the weighted synapse to the connected neurons.

Evolution constraints are as follows.

1. **Parallelism**: All neurons that meet the excitation condition execute their rules in synchrony.
2. **Priority**: When $p_i(t) = E_i$, only one rule can be selected and applied in each neuron.
3. **Halting condition**: The system halts at time $t$ if all neurons satisfy $p_i(t) \neq E_i$.

## A.2 ABLATION STUDY ON THE POTENTIAL ADJUSTMENT MODULE

This section validates the necessity of the potential adjustment module detailed in Section 3.3. Experimental results conducted on N-MNIST show that, without the module, the membrane potentials of neurons exhibit pronounced distributional collapse or drift over time, as follows.

**Distributional Collapse.** As shown in Table 7, when applying Xavier initialization, the membrane potentials collapse toward zero from the 30th to the 300th timestamp.

**Distributional Drift.** As shown in Table 7, when applying $U[-1, 1]$ initialization, the mean membrane potential drifts from $5.434$ to $51.327$ between the 30th and 300th timestamps, while the standard deviation increases from $8.143$ to $57.150$.

Additional experiments reveal that, without the potential adjustment module, the training accuracy of LSNP_IE remains around 10%, equivalent to random guessing, and the loss does not decrease. Therefore, given the fixed threshold $E_i$ and tolerance $\epsilon$, the narrow triggering interval $[E_i - \epsilon, E_i + \epsilon]$ in LSNP_IE hinders stable rule excitation. These findings highlight the necessity of incorporating the potential adjustment module to ensure effective and stable learning.

Furthermore, we validate the effects of $\mathcal{M}_1$. As shown in Table 8, without $\mathcal{M}_1$, the range of $p^{\text{in}}$ is mainly in $[-10, 16]$, while the range of $p^{\text{re}}$ only lies in $[0, 3]$, indicating a significant mismatch. This disparity causes the temporal features in $p^{\text{re}}$ to be overwhelmed during potential fusion. Table 8 shows that after applying $\mathcal{M}_1$, the distributions of $p^{\text{in}}$ and $p^{\text{re}}$ are well aligned, facilitating effective information integration across time steps.

Table 7: Distributional collapse and drift

| Timestamps | Collapse (Xavier) | | | | Drift ($U[-1,1]$) | | | |
|---|---|---|---|---|---|---|---|---|
| | **Mean** | **Std** | **Min** | **Max** | **Mean** | **Std** | **Min** | **Max** |
| 30 | 0.003 | 0.118 | -0.634 | 0.560 | 5.434 | 8.143 | -12.006 | 77.869 |
| 60 | -0.002 | 0.136 | -0.693 | 0.626 | 18.266 | 21.798 | -11.418 | 166.069 |
| 90 | 0.000 | 0.037 | -0.192 | 0.181 | 23.282 | 25.846 | -3.336 | 205.856 |
| 120 | 0.000 | 0.060 | -0.316 | 0.314 | 24.307 | 26.816 | -6.515 | 214.550 |
| 150 | 0.000 | 0.134 | -0.608 | 0.683 | 30.140 | 33.579 | -14.304 | 279.520 |
| 180 | -0.001 | 0.074 | -0.431 | 0.368 | 37.317 | 41.349 | -7.472 | 304.622 |
| 210 | 0.000 | 0.055 | -0.308 | 0.281 | 38.730 | 42.818 | -4.860 | 318.508 |
| 240 | 0.000 | 0.122 | -0.556 | 0.587 | 40.544 | 45.629 | -11.930 | 331.327 |
| 270 | 0.003 | 0.118 | -0.546 | 0.514 | 47.997 | 54.023 | -10.174 | 396.549 |
| 300 | 0.001 | 0.040 | -0.213 | 0.218 | 51.327 | 57.150 | -3.615 | 414.882 |

Table 8: Membrane potential ranges.

| Timestamps | Without $\mathcal{M}_1$ | | With $\mathcal{M}_1$ | |
|---|---|---|---|---|
| | $p^{\text{in}}$ | $p^{\text{re}}$ | $p^{\text{in}}$ | $p^{\text{re}}$ |
| 30 | $[-9.361, 17.645]$ | $[0, 3.354]$ | $[-0.521, 3.108]$ | $[0, 2.628]$ |
| 60 | $[-11.805, 18.433]$ | $[0, 3.241]$ | $[-0.472, 3.140]$ | $[0, 2.685]$ |
| 150 | $[-10.853, 17.198]$ | $[0, 3.145]$ | $[-0.636, 3.355]$ | $[0, 2.912]$ |
| 180 | $[-7.793, 14.842]$ | $[0, 3.596]$ | $[-0.866, 3.449]$ | $[0, 3.342]$ |
| 240 | $[-10.149, 12.651]$ | $[0, 3.477]$ | $[-0.540, 2.984]$ | $[0, 2.936]$ |
| 270 | $[-10.244, 16.150]$ | $[0, 3.075]$ | $[-0.444, 3.042]$ | $[0, 2.770]$ |

## A.3    SAMPLE IMAGES OF N-MNIST AND MNIST-DVS

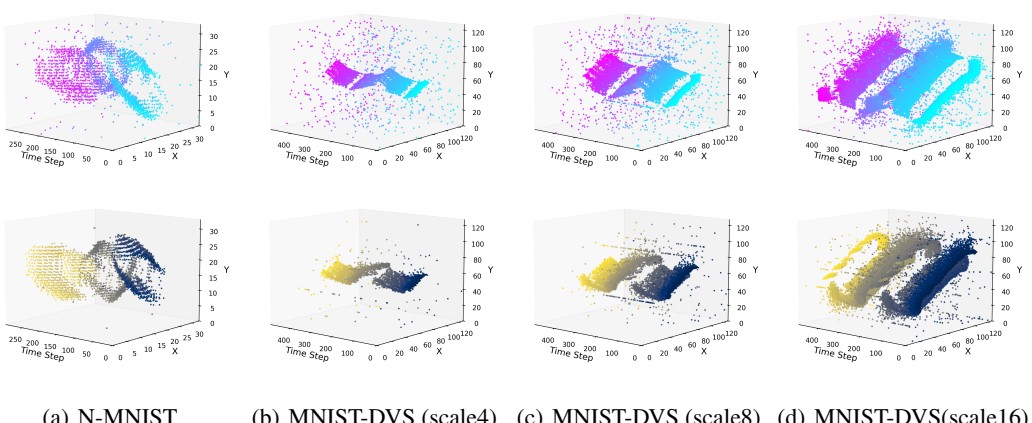

(a) N-MNIST     (b) MNIST-DVS (scale4)   (c) MNIST-DVS (scale8)  (d) MNIST-DVS(scale16)

Figure 4: Visualization of Samples (first row: ON events; second row: OFF events).

## A.4    USE OF LARGE LANGUAGE MODELS

During the preparation of this manuscript, we utilized a Large Language Model (LLM) as a writing assistant. The model's primary role was to assist in improving the clarity, conciseness, and overall readability of the text by refining sentence structures, polishing academic phrasing, and correcting grammatical errors. The LLM was not used for generating core research ideas, conducting experiments, analyzing results, or drawing the scientific conclusions presented in this paper. The authors have reviewed and edited all text and take full responsibility for the final content and scientific integrity of this work.

