# OpenReview forum: "Learnable Spiking Neural P System with Interval Excitation"
_ICLR.cc/2026/Conference — Submitted to ICLR 2026_

### Official Review · Reviewer_3yXf · 2025-10-28

**Soundness:** 2
**Presentation:** 3
**Contribution:** 2
**Rating:** 4
**Confidence:** 3

**Summary:**

This paper proposes a Learnable Spiking Neural P System with Interval Excitation (LSNP-IE) that integrates a differentiable learning framework into traditional Spiking Neural P (SN P) systems. The proposed LSNP-IE introduces three main innovations: (1) an interval excitation mechanism that replaces the point-triggered rule with a continuous interval, (2) a potential adjustment module to stabilize and normalize membrane potentials, and (3) a surrogate-gradient-based back-propagation algorithm for end-to-end training. Experiments on two neuromorphic datasets, N-MNIST and MNIST-DVS, show that LSNP-IE achieves competitive or superior accuracy compared to existing spiking and non-spiking baselines.

**Strengths:**

1. The paper extends the SN P system toward differentiable and adaptive learning.
2. The paper is well-structured with clear notation.
3. The presentation of back-propagation with surrogate gradients through interval excitation and potential adjustment is clear.

**Weaknesses:**

1. Evaluation is restricted to small-scale neuromorphic datasets (N-MNIST, MNIST-DVS). These are relatively simple and may not demonstrate scalability to complex or high-dimensional spatiotemporal tasks.
2. The paper does not report training/inference time, memory usage, or energy efficiency, which are central claims for spiking and membrane computing models.
3. LSNP-IE uses an MLP-style feed-forward topology only. The absence of convolutional or recurrent structures limits its comparison with advanced SNN architectures.
4. The study omits recent transformer-based or event-driven spiking frameworks.
5. Although the model is inspired by neural dynamics, the biological plausibility of the interval excitation mechanism is not discussed or experimentally supported.

**Questions:**

1. How does LSNP-IE scale to deeper or convolutional architectures? Could the interval excitation mechanism be integrated into spiking CNNs?

---

> ### Author Response · Authors · 2025-11-19
> **Response to Reviewer 3yXf**
>
> Thank Reviewer 3yXf for the insightful comments. We take responses to the doubts as follows.
>
> **W1: Evaluation is restricted to small-scale neuromorphic datasets**
>
> **A:** Initial N-MNIST/MNIST-DVS tests aimed to establish a baseline for this first learnable SN P framework on real-world tasks. Per your suggestion, we added experiments on the challenging DVS-Gesture and N-TIDIGITS datasets:
>
> | **Dataset** | **Model**      | **Architecture** | **Accuracy (%)** |
> | ----------- | -------------- | ---------------- | ---------------- |
> | DVS-Gesture | LSNP_IE (Ours) | MLP              | **84.94**        |
> |             | R. Iyer’s [1]  | FC-SNN           | 53.18            |
> |             | Zhang’s [2]    | FC-SNN           | 84.76            |
> |             | Zhang’s [3]    | FC-SNN           | 83.50            |
> | N-TIDIGITS  | LSNP_IE (Ours) | MLP              | **65.97**        |
>
> Results confirm LSNP_IE generalizes well, even cross-modally (vision to audio). To our knowledge, this is the first SN P system applied to these benchmarks, achieving performance comparable to mature SNNs. Notably, our 84.94% accuracy on DVS-Gesture outperforms several SNN baselines [1,2,3]. While N-TIDIGITS performance (65.97%) trails SOTA SNNs, we attribute this to the MLP's inherent limitations in capturing complex temporal features, not the LSNP_IE mechanism. This recognition inspires us to explore SN P systems with various architectures in the future.
>
> ---
>
> **W2: The paper does not report training/inference time, memory usage, or energy efficiency**
>
> **A:** We summarize computational resources and efficiency below:
>
> | **Metric**           | **N-MNIST**                    | **MNIST-DVS**                 |
> | -------------------- | ------------------------------ | ----------------------------- |
> | **Platform**         | RTX 4090 (24GB VRAM)           | RTX 4090 (24GB VRAM)          |
> | **Training Time**    | ~ 4 hours                      | ~ 4 hours                     |
> | **Inference Time**   | ~ 1 minute                     | ~ 3 minutes                   |
> | **Avg. SOPs/Sample** | 37.6 M (vs. 423.3 M Dense Ops) | 73.7 M (vs. 8.32 B Dense Ops) |
>
> Note that standard FLOPs/SOPs do not fully reflect true SN P efficiency, as SN P systems are inherently designed for distributed, parallel hardware, efficiency stems from structural modularity and event-driven dynamics rather than centralized computation[4].
>
> ---
>
> **W3, Q1: LSNP-IE uses an MLP-style feed-forward topology only, lacking CNN or RNN structures, which limits its comparison with advanced SNN architectures**
>
> **A:** Our work's primary goal is investigate how to make SN P systems differentiable and capable of end-to-end optimization, rather than to outperform SOTA SNNs on classification tasks. We chose MLP as the clearest validation platform; if training fails on a simple MLP, discussing complex architectures is moot.
>
> Following your suggestion, we tested a convolutional front-end (LSNP-CNN), achieving 82.32% (vs. 84.94% for LSNP-MLP). This indicates adapting CNNs is non-trivial; simple layer replacement yields no gain and requires deep tuning of the interaction between convolution and SN P dynamics—a key focus for our future work.
>
> ---
>
> **W4: The study omits recent transformer-based or event-driven spiking frameworks**
>
> **A:** We acknowledge Transformer-based SNNs as cutting-edge. However, as mentioned, this study focuses on building a foundational differentiable framework for SN P systems. Comparing the newly proposed LSNP_IE directly with highly optimized SOTA SNN architectures is premature. We believe that integrating LSNP_IE with Transformers is a relatively significant but next-step research direction.
>
> ---
>
> **W5: The biological plausibility of the interval excitation mechanism is not discussed or experimentally supported**
>
> **A:** Unlike traditional SNNs focusing on biophysical simulation, the core value of SN P systems lies in their hardware-friendly parallel computing paradigm, not strict biological plausibility. "Interval excitation" is a mathematical abstraction designed to resolve non-differentiability in continuous domains. This prioritizes computational stability to unlock potential for edge/FPGA implementation rather than mimicking biological constraints.
>
> ---
>
> **References**
>
> [1] Is neuromorphic mnist neuromorphic? analyzing the discriminative power of neuromorphic datasets in the time domain
>
> [2] Self-backpropagation of synaptic modifications elevates the efficiency of spiking and artificial neural networks
>
> [3] Self-lateral propagation elevates synaptic modifications in spiking neural networks for the efficient spatial and temporal classification
>
> [4] Improving GPU web simulations of spiking neural P systems

---

### Official Review · Reviewer_xReB · 2025-10-28

**Soundness:** 3
**Presentation:** 2
**Contribution:** 2
**Rating:** 4
**Confidence:** 3

**Summary:**

This paper introduces a model named "Learnable Spiking Neural P System with Interval Excitation" (LSNP IE), which aims to address two core challenges of traditional Spiking Neural P (SN P) systems when processing real-world data: their limited expressive capacity and the non-differentiable nature of their excitation mechanism. The paper tries to discuss three key innovations:

1. Interval Excitation Mechanism: The traditionally strict point-triggered firing condition (where the potential must exactly equal the threshold) is relaxed to an interval. This improves the model's robustness and ensures continuous information flow when handling real-valued (floating-point) data.
2. Potential Adjustment Module: The paper introduces two normalization-like modules (M1 and M2) to align the input and residual potentials and to shift the fused potential's distribution towards the excitation interval, ensuring firing stability and effective learning.
3. Surrogate Gradient-based End-to-End Training: The Surrogate Gradient (SG) method is employed to handle the non-differentiable parts of the firing function, enabling end-to-end backpropagation-based training for the entire network.

The authors validate their method on neuromorphic datasets like N-MNIST and MNIST-DVS, reporting competitive performance compared to traditional non-spiking and spiking models. The main contribution of this work lies in bridging the gap between the theoretically-oriented SN P systems and modern deep learning training frameworks, demonstrating the feasibility of applying such models to vision tasks.

**Strengths:**

1. Problem-Driven Design. The proposed "interval excitation" and "potential adjustment" modules are well-thought-out. The interval excitation directly addresses the issue of point-triggered firing having near-zero probability in a continuous domain. The potential adjustment module counters the "distributional collapse" or "distributional drift" issues during training, which is crucial for stable learning. The effectiveness of these modules is also validated through ablation studies.
2. Clear Structure. The paper is well-structured, logically flowing from background to method, experiments, and conclusion. The appendix provides a detailed introduction to WSN P systems and supplementary experiments on the necessity of the potential adjustment module, all of which strongly support the reader's understanding of the paper's core ideas.

**Weaknesses:**

1. For the broader audience, SN P systems are a relatively niche concept. The introduction and related work mention that SN P systems possess a "parallel and distributed architecture" and "modularity," hinting at their potential for hardware implementation. However, the paper fails to articulate more concretely what unique advantages this architecture offers over more established SNNs or traditional ANNs for tasks like image classification. This lack of justification might leave readers questioning the necessity of introducing the complexity of P systems.
2. The potential adjustment module is formally very similar to Batch Normalization or Layer Normalization. While the ablation study proves its necessity, the paper could provide a deeper discussion on its design motivation. Is it merely an engineering trick to "push" the potential values into the firing interval? Did the authors consider other, simpler methods (like clipping or simpler scaling/shifting)? Clarifying this would help in understanding whether this module is a general-purpose solution within the SN P framework.
3. The paper is somehow a disconnect between its core motivation and its experimental design. It argues that SN P systems offer advantages beyond standard SNNs, but then evaluates the model exclusively on image classification—a task where SNNs are already well-established and highly effective. I think the experiments thus fail to provide a compelling answer to the crucial concern: why should one utilize SN P framework if they do not showcase a experimentally unique capability (e.g., in computational modeling, structured problem-solving) that would justify its additional complexity over SNNs.

**Questions:**

1. About the Motivation of SN P Systems: could the authors elaborate on what practical or potential advantages (e.g., in computation, energy efficiency, or scalability) the parallel and modular structure of SN P systems offers for a task like image classification, compared to standard SNNs?
2. Could you comment on the choice of experimental tasks? Given that a key motivation for SN P systems is their unique structure inherited from P systems, have you considered tasks where this structure might offer a more distinct advantage over standard SNN architectures (beyond visual classifications)? Furthermore, i think provide a comparison or discussion regarding more recent SOTA SNNs on these datasets could be better to contextualize performance of the proposed method.

---

> ### Author Response · Authors · 2025-11-19
> **Response to Reviewer xReB**
>
> We sincerely appreciate Reviewer xReB’s insightful suggestions. Below are our detailed responses to each concern:
>
> **W1, Q1: On the unique advantages of SN P systems and the necessity of introducing them**
>
> **A:** We fully agree with the reviewer's perspective; this is the most central issue that needs clarification in our work. We must re-frame the motivation of our work: Solving the "Trainability" Bottleneck of SN P Systems. Before our work, SN P systems (despite being Turing-complete in theory) heavily relied on experts to manually design rules, making them impossible to scale to any large-scale, real-world problems (like N-MNIST). The primary contribution of this paper is not to outperform SNNs on image classification, but to be the first to provide a differentiable, end-to-end trainable framework for the SN P computational paradigm, thus breaking the critical barrier between theory and practice.
>
> About the Unique Advantages of SN P Systems. Firstly, Isomorphism between Computation and Hardware: The advantage of SNNs lies in "biological simulation," while the advantage of SN P systems lies in "computation abstraction." SN P systems, derived from P systems (membrane computing), have a "membrane-rule-object" structure that is, in theory, more isomorphic to parallel, distributed, and modular hardware design paradigms than the "neuron-synapse" model of SNNs. It is precisely this hardware affinity that gives SN P systems great potential for implementing highly energy-efficient, low-power dedicated hardware.
>
> Secondly, Positioning of This Paper: Our work is the "software key" to unlock this potential. Before they could be effectively trained, the hardware advantages of SN P systems were purely theoretical ("castles in the air"). By enabling SN P models to be trained on standard hardware (GPUs) via LSNP_IE, we provide the first step toward their future deployment on specialized neuromorphic hardware.
>
> About Why Choose Image Classification Tasks? We have to claim that we chose a considerably mature neuromorphic computing task, rather than an image classification task.
>
> ---
>
> **W2: On the design motivation of the Potential Adjustment Module**
>
> **A:** We deeply appreciate the reviewer's technical question.
>
> The reviewer's observation is very accurate; it does formally borrow from BN/LN. However, its motivation is not a simple "engineering trick" but a necessity to solve the problem of "potential distribution collapse or drift" when SN P systems process continuous floating-point numbers over many timesteps. As shown in Table 7 of Appendix A.2, without this module, the neuron potentials rapidly either "collapse" (mean to zero) or "drift" (mean grows unboundedly) over time, making it impossible for potentials to stably fall within the activation interval and thus causing a breakdown of information flow.
>
> We did consider Clipping, but it can only "truncate" outliers; it cannot pull an entire drifted distribution back into the activation interval and thus cannot solve the problem at its root. In contrast, our potential adjustment module (functionally similar to BN) adjusts the potential distribution at every single timestep, forcibly "pushing" it back near the activation interval.
>
> In summary, the potential adjustment module is a core component we customized for SN P systems to maintain the stability of information flow and the continuity of gradients, and it is an essential prerequisite for making deep SN P networks trainable.
>
> ---
>
> **W3, Q2: On the choice of experimental tasks and comparison with SOTA SNNs**
>
> **A:** Thank you for your valuable suggestions; this is closely related to the core issue in Q1.
>
> - **On Task Choice (Image Classification vs. Unique Tasks)**: As stated in Q1, we chose image classification as a benchmark validation, which is a necessary step to prove a new framework's effectiveness. Besides, we completely agree with the reviewer that the unique structure of SN P systems (from P systems) gives them great potential for tasks beyond visual classification.
> - **On Comparison with SOTA SNNs**: We acknowledge the lack of SOTA SNN comparisons in Table 2. We will add some SOTA SNN results in the revised version to better contextualize our model's performance. However, we must also emphasize that the LSNP_IE proposed in this paper is based on an MLP architecture, whereas many SOTA SNNs are based on complex CNN or Transformer architectures. Therefore, a direct accuracy comparison suffers from an architectural unfairness (an "apples-to-oranges" comparison). Our comparison with SOTA SNNs is not intended to "surpass" them, but to "prove feasibility"—that is, our entirely new MLP framework designed for SN P systems can achieve performance competitive with MLP/CNN baselines optimized for SNNs, which fully demonstrates our framework's effectiveness and potential.

---

### Official Review · Reviewer_rKMm · 2025-10-29

**Soundness:** 3
**Presentation:** 3
**Contribution:** 3
**Rating:** 4
**Confidence:** 2

**Summary:**

This paper proposes a Learnable Spiking Neural P System(LSNP_IE) to improve the expressive capacity of traditional SN P systems, using the interval excitation mechanism and potential adjustment module. The authors introduce the surrogate gradients, which are widely used in SNNs, to train the SN P system. The performance of LSNP_IE is evaluated on two neuromorphic datasets, N-MNIST and MNIST-DVS.

**Strengths:**

1.The paper proposes an innovative SN P system.
2.The paper has a complete writing structure and clear logic.

**Weaknesses:**

1.The experiments lack validation on higher-resolution datasets.
2.The paper lacks comparative studies with other SN P systems of the same type.
3.The hyperparameter sensitivity analysis is incomplete.
4.The experimental results report the standard deviation, but the experimental settings do not specify which random number seeds are used.

**Questions:**

1.Why does LSNP_IE perform significantly worse on the scale 16 of the MNIST-DVS dataset compared to the other two scales?
2.Why are tests not conducted on static images? Can LSNP_IE only be applied to neuromorphic data?
3.Why is it difficult for existing SN P systems to adopt complex structures such as convolutional layers?

---

> ### Author Response · Authors · 2025-11-19
> **Response to Reviewer rKMm**
>
> Thank Reviewer rKMm for insightful comments. We take responses to the doubts as follows.
>
> **W1: Lack of Validation on Higher-Resolution Datasets**
>
> **A:** Initial N-MNIST/MNIST-DVS tests aimed to establish a baseline for this first learnable SN P framework on real-world tasks. Per your suggestion, we added experiments on the challenging DVS-Gesture and N-TIDIGITS datasets:
>
> | **Dataset** | **Model**      | **Architecture** | **Accuracy (%)** |
> | ----------- | -------------- | ---------------- | ---------------- |
> | DVS-Gesture | LSNP_IE (Ours) | MLP              | **84.94**        |
> |             | R. Iyer’s [1]  | FC-SNN           | 53.18            |
> |             | Zhang’s [2]    | FC-SNN           | 84.76            |
> |             | Zhang’s [3]    | FC-SNN           | 83.50            |
> | N-TIDIGITS  | LSNP_IE (Ours) | MLP              | **65.97**        |
>
> Results confirm LSNP_IE generalizes well, even cross-modally (vision to audio). To our knowledge, this is the first SN P system applied to these benchmarks, achieving performance comparable to mature SNNs. Notably, our 84.94% accuracy on DVS-Gesture outperforms several SNN baselines [1,2,3]. While N-TIDIGITS performance (65.97%) trails SOTA SNNs, we attribute this to the MLP's inherent limitations in capturing complex temporal features, not the LSNP_IE mechanism. This recognition inspires us to explore SN P systems with various architectures in the future.
>
> ---
>
> **W2: Lack of Comparison with Same-Type SN P Systems**
>
> **A:** We fully agree that comparing our work with models of the same type is crucial. However, the fundamental reason we primarily compared against SNN models in the paper is that there are currently almost no other public, trainable SN P system benchmarks capable of handling large-scale neuromorphic datasets (like N-MNIST). Traditional SN P systems heavily rely on experts manually constructing rules[4], which makes them difficult to scale to complex learning tasks.
>
> ---
>
> **W3: The hyperparameter sensitivity analysis in the paper is incomplete**
>
> **A:** We acknowledge that, due to space limitations, our original ablation studies (Section 4.3) prioritized the components most critical to model performance (such as weight initialization, potential decay coefficient $d$, and choice of surrogate gradient). To make our study more rigorous, we conducted analysis experiments on the N-MNIST dataset during the rebuttal period for the threshold tolerance $\epsilon$ and the number of time frames.
>
> **Threshold Tolerance** $\epsilon$**:**
>
> | **Tolerance ϵ**  | **0.2** | **0.25** | **0.3 (Ours)** | **0.35** | **0.4** |
> | ---------------- | ------- | -------- | -------------- | -------- | ------- |
> | **Accuracy (%)** | 94.20   | 97.24    | **97.91**      | 97.53    | 95.44   |
>
> The experimental results show a clear "inverted U-shaped" trend, confirming that $\epsilon$ is a key parameter for balancing the model's selectivity and information flow: a too-narrow interval ($\epsilon=0.2$**)** causes valid signals to be filtered out, thus hindering information propagation, while a too-wide interval ($\epsilon=0.4$**)** introduces noise and leads to unstable potential dynamics. Our setting of $\epsilon=0.3$ strikes the right balance between signal sensitivity and noise suppression, thereby achieving optimal performance.
>
> **Number of Time Frames:**
>
> | Time Frames      | 240   | 260   | 280   | 300 (Ours) | 320   | 340   | 360   |
> | ---------------- | ----- | ----- | ----- | ---------- | ----- | ----- | ----- |
> | **Accuracy (%)** | 97.75 | 97.71 | 97.85 | **97.91**  | 97.74 | 97.84 | 97.79 |
>
> The experimental results show that, holding other parameters constant, changing the number of time frames does not have a significant impact on accuracy. This demonstrates the stability of the SN P system parameters we used. Among all settings, 300 time frames achieved the best result, suggesting it captures the information most effectively.
>
>
> ---
>
> **References**
>
> [1] Is neuromorphic mnist neuromorphic? analyzing the discriminative power of neuromorphic datasets in the time domain
>
> [2] Self-backpropagation of synaptic modifications elevates the efficiency of spiking and artificial neural networks
>
> [3] Self-lateral propagation elevates synaptic modifications in spiking neural networks for the efficient spatial and temporal classification
>
> [4] Spiking neural P systems
>
> (Response continues in the next comment due to character limits...)

---

> ### Author Response · Authors · 2025-11-19
> **Response to Reviewer rKMm**
>
> (Continued from previous comment)
>
> **W4: The experimental results report the standard deviation, but the experimental settings do not specify which random number seeds are used**
>
> **A:** We sincerely apologize for the insufficient description in Section 4.1 of our paper and hereby clarify that all reported means and standard deviations (e.g., 97.91±0.04% on N-MNIST) are based on results from five independent runs. To ensure reproducibility, the experiments were conducted under fixed data splits and batch generation (with random_seed=42), guaranteeing determinism for each individual run. The five independent runs were achieved by varying other random seeds (e.g., for weight initialization).
>
> We commit to releasing all code and experimental scripts immediately upon acceptance of the paper to ensure full reproducibility of the results.
>
> ---
>
> **Q1: Regarding the reasons for performance degradation on Scale 16 of the MNIST-DVS dataset**
>
> **A:** This is a key observation. First, it should be clarified that according to the original dataset papers [5], "scale" refers to the spatial magnification factor applied to the original MNIST digits (as shown in Figure 4 of Appendix A.3), with scale 16 being the largest and most spatially expanded version.
>
> The root of the problem lies in the limitations of the MLP architecture: it cannot effectively capture spatial locality like CNNs do. When the digits are small (scale 4/8), event signals are concentrated, and the MLP can still process them reasonably well. However, when the digits are enlarged to scale 16, structural components become too distant in the flattened high-dimensional vector, making it difficult for the MLP to capture such long-range spatial dependencies.
>
> Therefore, we argue that the performance drop on scale 16 is not caused by the LSNP_IE mechanism itself, but rather stems from the MLP's inherent difficulty in handling large-scale spatial features after flattening.
>
> ---
>
> **Q2: Why not test on static images, and can LSNP_IE only be applied to neuromorphic data?**
>
> **A:** Thank you for the question. Our LSNP_IE framework is fully capable of processing static images. A common-used practice is to encode static images (like MNIST) over time (e.g., rate coding or Poisson coding), converting them into spike sequences, which can then be fed into our model.
>
> We chose to focus on neuromorphic datasets (N-MNIST, MNIST-DVS) in this paper for two main reasons. The first is the **Data Naitivity**. Neuromorphic data (event streams) are the native input format for spiking models (both SNNs and SN P systems). They inherently possess spatiotemporal properties that best demonstrate the advantages of event-driven computation. The second is the **Research Motivation**. A core motivation for SN P systems and SNNs is to emulate the high energy efficiency and event-driven nature of biological neural systems. Using native neuromorphic data is the most direct validation of this motivation.
>
> Therefore, our choice of neuromorphic data was to better match the "spiking" nature of the model, rather than a limitation of the model's capability.
>
> ---
>
> **Q3: Why is it difficult for existing SN P systems to adopt complex structures such as convolutional layers?**
>
> **A:** Previously, SN P systems lacked a differentiable training framework, making even basic structures like MLPs difficult to train. Our core contribution establishes this foundation, paving the way for complex architectures such as CNNs.
>
> Following your suggestion, we tested a convolutional front-end (LSNP-CNN), achieving an accuracy of 82.32% (compared to 84.94% for LSNP-MLP). This indicates that successfully adapting CNNs to the SN P framework is non-trivial; simple layer replacement does not yield immediate performance gains and requires deep tuning of the interaction between convolution operations and SN P dynamics—a compelling direction for our future work.
>
> ---
>
> **References**
>
> [5] Poker-DVS and MNIST-DVS. Their history, how they were made, and other details

---

### Official Review · Reviewer_8hy1 · 2025-11-01

**Soundness:** 3
**Presentation:** 3
**Contribution:** 3
**Rating:** 4
**Confidence:** 3

**Summary:**

This paper introduces a Learnable Spiking Neural P (LSNP_IE) system, contributing a novel interval excitation mechanism that enables effective gradient-based training for SN P systems on neuromorphic datasets. It provides a mathematically precise framework and demonstrates competitive performance, representing a step towards bridging the theory of membrane computing with practical learning. However, the work is limited by its use of a simple MLP architecture, which fails to demonstrate scalability, and provides no empirical evidence for the core claimed advantage of energy efficiency, leaving its practical superiority over established spiking neural models unproven.

**Strengths:**

1.	It introduces a learnable Interval Excitation Mechanism, effectively solving the "probability zero" firing problem for SN P systems with continuous data, a significant conceptual advance in the field.
2.	It demonstrates end-to-end gradient-based training for an SN P system, a crucial step towards bridging the gap between their theoretical potential and practical application on real-world tasks.
3.	It provides a mathematically precise definition of LSNP_IE and supports it with thorough ablation studies (e.g., on surrogate gradients, decay coefficient d) that empirically validate design choices.

**Weaknesses:**

1.	The core innovation, the interval excitation mechanism, is essentially a relaxation of a discrete threshold to a continuous one—a well-established concept in traditional SNNs (e.g., the use of surrogate gradients often implicitly does this). The potential adjustment module is a direct analog of Batch Normalization, adapted for membrane potentials. While the application of these ideas to the SN P system formalism is new, the conceptual building blocks are largely borrowed from adjacent fields. The paper does not demonstrate a fundamental theoretical advance in membrane computing itself.
2.	Using a simple Multi-Layer Perceptron (MLP) is outdated and fails to demonstrate the model's compatibility with modern deep learning architectures. The claim that the framework supports "arbitrary depth" is unsupported, as no deep or convolutional networks are tested. This raises serious doubts about its practical applicability.
3.	A primary motivation for SN P systems is their purported energy efficiency on neuromorphic hardware. However, the paper provides zero evidence for this claim. There is no analysis of computational complexity, number of spikes, or energy consumption compared to standard SNN baselines. Without this, the work remains a purely theoretical exercise, and its advantage over simpler, more established SNN models is unproven.
4.	Given that the interval excitation is conceptually similar to the soft-thresholding used in surrogate gradient methods for SNNs, what is the fundamental advantage of the LSNP_IE formalism over a standard, well-optimized SNN with surrogate gradients? The experiments currently show comparable, not superior, performance on simple tasks.
5.	Lack of Empirical Evidence for Computational Efficiency Claims: A core motivation for spiking models and neuromorphic hardware is energy efficiency. The paper repeatedly mentions this advantage (e.g., "significant energy efficiency advantages," "low-energy implementations") but provides no empirical measurements or estimates to support this claim for LSNP_IE. There is no analysis of the number of synaptic operations (SOPs), spike counts, or any other proxy for energy consumption compared to the baseline models. Without this data, the practical benefit of LSNP_IE for low-power applications remains an unverified assertion.

**Questions:**

The paper emphasizes the 'inherently parallel, distributed, and modular structure' of SN P systems as a key differentiator from 'monolithic SNNs.' However, in the presented LSNP_IE, which uses a standard MLP topology, how does this modularity manifest in a way that is functionally different from a standard, layered SNN? Could you specify a concrete computational or representational benefit provided by the membrane/rule formalism in this specific instantiation that cannot be achieved by an SNN?

---

> ### Author Response · Authors · 2025-11-19
> **Response to Reviewer 8hy1**
>
> We sincerely thank Reviewer 8hy1 for invaluable comments. Below are our detailed responses to each concern:
>
> **W1: On Core Innovation, "Borrowed" Concepts, and Advance in Membrane Computing**
>
> **A:** We acknowledge that these mechanisms are conceptually similar to SNN/ANN techniques. However, our core innovation lies in non-trivially adapting them to SN P systems to solve fundamental obstacles. The SNP system proposed in this paper effectively promotes the solution of problems in this field, which is obviously more important for developing SN P systems.
>
> 1. **Necessity of "Interval Excitation":** Traditional SN P systems rely on a strict "point-trigger" mechanism (i.e., potential $p _i$ must exactly equal $E _i$). When processing real-world floating-point potentials, this causes information flow interruption. Our "interval excitation" is a fundamental relaxation of this mechanism, enabling SN P systems to process continuous potential values for the first time.
> 2. **Necessity of "Potential Adjustment":** As shown in Appendix A.2, without this module, membrane potential distributions suffer from severe "collapse or drift." Since potentials easily drift out of the active interval over time steps, the network stops firing, interrupting information flow. Therefore, this module is not merely Batch Normalization, but a "temporal potential stability component" for SN P systems.
>
> ---
>
> **W2: On the limitations of the MLP architecture and the absence of CNNs**
>
> **A:** We acknowledge that the MLP is the simple but common-used architecture. Our primary goal is to prove the "trainability" of SN P systems. Therefore, validating the effectiveness and stability of our novel framework on the clearest benchmark platform (i.e., MLP on N-MNIST) is a necessary first step. Extending this to more architectures is something ones need to do in the future.
>
> Following your suggestion, we tested a convolutional front-end (LSNP-CNN) on DVS-Gesture, achieving an accuracy of 82.32% (compared to 84.94% for our LSNP-MLP). This result proves that adapting LSNP_IE to deeper CNN architectures is a non-trivial research direction. Simple layer replacement does not work "out-of-the-box" and requires deeper tuning of the interplay between convolution and SN P dynamics. As noted in our conclusion, exploring deeper and more complex architectures (like CNNs) in practice is a key focus of our future work.
>
> ---
>
> **W3, W5: On the lack of empirical evidence for energy efficiency claims**
>
> **A:** We must clarify that this paper focuses on developing a trainable SN P software model. Before our work, training SN P models on large-scale datasets like N-MNIST was impossible, making efficiency analysis unfeasible. We acknowledge that we have not yet verified these advantages through hardware deployment, and we will pursue hardware-level testing in subsequent work.
>
> Furthermore, to provide empirical evidence, we conducted a supplementary analysis of Synaptic Operations (SOPs) during inference. We calculated the ratio of actual spike-triggered accumulations to standard ANN theoretical operations, providing strong support for the potential of low-energy implementation.
>
> | **Dataset** | **SOPs (Ours)** | **Dense Ops (ANN)** | **Ratio**   |
> | ----------- | --------------- | ------------------- | ----------- |
> | N-MNIST     | 37.6 M          | 423.3 M             | ~ **1/11**  |
> | MNIST-DVS   | 73.7 M          | 8.32 B              | ~ **1/112** |
>
> ---
>
> **W4: On the fundamental advantage of LSNP_IE over SNNs**
>
> **A:** We frankly acknowledge that LSNP_IE shows no direct performance advantage on current image classification tasks. However, our goal is not to outperform SNNs in their most mature domain, but to "unlock" a trainable SN P system. This enables leveraging SN P's unique advantages in large-scale tasks for the first time.
>
> SNNs excel at "biological simulation," whereas SN P systems excel at "computation abstraction". The "membrane-rule" structure of SN P systems is theoretically more isomorphic to parallel, distributed hardware than the "neuron-synapse" model of SNNs. Our work is the "software key" to unlock this potential, enabling training on standard hardware (GPUs).
>
> ---
>
> **Q1: On the "modularity" advantage of SN P systems within an MLP topology**
>
> **A:** We thank the reviewer for this profound question. The current specificity of SN P lies in its hardware-friendly potential. Specifically, their inherent modularity makes them particularly suitable for edge deployment and FPGA implementation. The "separation of container and logic" paradigm offers potential difficult for traditional SNNs to achieve:
>
> - **Heterogeneous Computing:** Allowing mixed learnable rules (perception) and symbolic logic rules (reasoning) in one system.
> - **Hardware Modularity:** Membranes can be implemented as standardized memory units, and rules as pluggable logic units. This unlocks a unique potential for neuro-symbolic computation via LSNP_IE.

---

### Meta-Review · Area_Chair_53Xg · 2026-01-07

**Summary:**

The paper proposes an interval-excitation–based learning framework (LSNP-IE) for SN P systems. The main concerns regarding this work can be summarized as follows:

1. The core ideas show limited conceptual novelty. Interval excitation is essentially a soft or continuous threshold mechanism long used in SNNs, and the proposed potential adjustment module closely resembles Batch Normalization applied to membrane potentials. While their application to the SN P formalism is new, the work does not deliver a fundamental theoretical advance in membrane computing itself.
2. Empirically, the evaluation is weak and outdated. Experiments rely on shallow MLP-style networks and small neuromorphic datasets, with no demonstration of scalability to deeper, convolutional, recurrent, or modern event-driven architectures, despite claims of arbitrary depth.
3. Core motivations such as energy efficiency and suitability for neuromorphic hardware are not substantiated. There is no analysis of spike counts, SOPs, computational complexity, or energy consumption is provided. Comparative studies with other SN P systems and strong SNN baselines are missing, hyperparameter sensitivity and experimental reproducibility are insufficiently documented, and the biological plausibility of the interval-excitation mechanism is not convincingly justified.
4. Finally, there is a disconnect between the motivation for SN P systems and the chosen evaluation tasks, which do not showcase capabilities that would clearly justify the added complexity over established SNN frameworks.

**Reviewer Concerns:**

While the authors provide extensive responses and make a clear effort to defend the novelty and motivation of LSNP-IE, the rebuttal remains largely reframing rather than resolving the core issues. Many responses rely on asserting necessity or future potential without providing concrete theoretical justification or decisive empirical evidence. Key concerns such as limited conceptual novelty, lack of compelling advantages over standard SNNs, insufficient validation on modern architectures, and unsubstantiated claims about hardware benefits, remain only partially addressed. Additional experiments introduced in the rebuttal are still restricted in scale and do not fundamentally change the conclusions drawn from the original evaluation. As a result, despite the authors’ clarifications and added analyses, the rebuttal does not convincingly address the major concerns raised by reviewers.

**Reviewer Scores:**

All four reviewers assigned borderline reject scores, reflecting shared concerns despite some interest in the topic. While the authors’ rebuttal offers clarifications and additional experiments, it mainly reframes the contribution and does not resolve most of the fundamental issues raised across reviews, including limited novelty, unclear advantages over established SNN frameworks, and insufficient empirical evidence for scalability. I believe that only the concerns raised by Reviewer rKMm were largely and convincingly addressed. For the remaining reviewers, it is unlikely that their scores would have increased had they participated fully in the discussion.

---

### Decision · Program_Chairs · 2026-01-26

Reject